# Effect of pH Hydrolysis on the Recovery of Antimony from Spent Electrolytes from Copper Production

**DOI:** 10.3390/ma16113918

**Published:** 2023-05-23

**Authors:** Eduardo Díaz Gutiérrez, José Antonio Maldonado Calvo, José María Gallardo Fuentes, Antonio Paúl Escolano

**Affiliations:** 1Departamento de Ingeniería y Ciencia de los Materiales y del Transporte, Escuela Politécnica Superior, Calle Virgen de África 7, 41011 Seville, Spain; josemar@us.es (J.M.G.F.); apaul@us.es (A.P.E.); 2Atlantic Copper, S.L.U., Francisco Montenegro Avenue, 21001 Huelva, Spain; jmaldona@fmi.com

**Keywords:** antimony metallurgy, antimony oxychloride, hydrolysis, copper electrorefining

## Abstract

This study examined how pH hydrolysis affects the recovery process for antimony extracted from spent electrolytes. Various OH^−^ reagents were used to adjust the pH levels. The findings reveal that pH plays a crucial role in determining the optimal conditions for extracting antimony. The results show that NH_4_OH and NaOH are more effective compared to water, with optimal conditions at pH 0.5 for water and pH 1 for NH_4_OH and NaOH, resulting in average antimony extraction yields of 90.4%, 96.1%, and 96.7%, respectively. Furthermore, this approach helps to improve both crystallography and purity related to recovered antimony samples obtained through recycling processes. The solid precipitates obtained lack a crystalline structure, making it difficult to identify the compounds formed, but element concentrations suggest the presence of oxychloride or oxide compounds. Arsenic is incorporated into all solids, affecting the purity of the product, and water showing higher antimony content (68.38%) and lower arsenic values (8%) compared to NaOH and NH_4_OH. Bismuth integration into solids is less than arsenic (less than 2%) and remains unaffected by pH levels except in tests with water, where a bismuth hydrolysis product is identified at pH 1, accounting for the observed reduction in antimony extraction yields.

## 1. Introduction

The concept of circular economy (CE) arises from two fundamental needs. First, to try to reduce the consumption of raw materials. Second, to reduce the production of waste which, in turn, is connected to both economic and environmental factors. This concept has traditionally been linked to the reintroduction or reprocessing of materials or elements once the useful life of the product or application they are part of is finished. However, in the field of extractive metallurgy and mineral processing, this concept can be applied to the recovery of any material that may be of interest in different phases of the production process in which streams are generated and are often treated as waste [1].

In this concern, the European Union (EU) established a specific policy on critical raw materials in 2008 [2], which focuses on ensuring sustainable and secure access to raw materials. Some of the policies and actions that the EU has implemented include: (I) identifying critical raw materials based on the assessment of factors such as supply risk, economic importance, and environmental impact; (II) promoting the circular economy and material recovery to reduce the need for new raw materials and reduce dependence on critical raw materials; (III) investing in research and innovation to develop alternatives to critical raw materials, as well as improve efficiency in their use. 

Aligned with these objectives and in the search for candidate processes, during the production of high-purity copper a wide variety of materials are generated that have traditionally been treated as waste but can be converted into recyclable materials or by-products that can be marketed or used in other sectors as raw materials through certain actions. Specifically, the stream generated in the antimony and bismuth elimination plant of the electrolyte used in the final electrorefining process of copper at the Atlantic Copper metallurgical complex in Huelva turns out to be a candidate source. Some reasons support this issue. First, antimony is qualified as a critical raw material by the European Union [3,4]. Second, currently, the stream is neutralized and treated as waste. Finally, the metals are found in a liquid stream of hydrochloric acid that allows the application of hydrometallurgical methods. 

The by-product stream is produced during the regeneration stage of the chelating resins used to remove antimony and bismuth from the copper electrolyte [5]. At this point, in a process known as elution, the resulting solution or discharge is commonly referred to as an eluate. It consists of an hydrochloric acid stream with chloride concentrations of approximately 170 g/L and a varying content of Sb and Bi [6]. Table 1 displays the standard composition of the eluate.

Antimony is a chemical element that has been used in a variety of applications, including as a component of alloys, in flame retardants, and in medicines [7,8]. Recently, new uses of antimony have emerged, particularly in the fields of energy storage and electronics. Antimony is being investigated as a potential anode material for lithium-ion batteries [9], which are used in a wide range of devices from smartphones to electric vehicles. Antimony-based materials are also being explored for use in high-performance electronic devices, such as transistors and diodes, due to their unique electrical and optical properties [10]. Additionally, antimony could be an important material in the development of new solar cell technologies, particularly in the development of thin-film solar cells [11,12]

In this context, several authors [13,14,15] have published reviews about the current technological situation of antimony recovery, covering the current market and resource situation, both primary and secondary sources, and finally, the existing procedures for production and recovery depending on the medium. Within the different hydrometallurgical processes described in the reviews, hydrolysis appears as one of the procedures applied for the recovery of antimony in acidic streams.

Indeed, hydrolysis is described as one of the procedures of the hydrometallurgy method for industrial production of antimony [16], where antimony in sulfide form is transformed into chloride by leaching it in a chloride medium, reaction (1). The antimony chloride is then obtained in a metallic form by electrowinning in diaphragm cells. Antimony chloride can also be extracted by hydrolysis by adding OH^−^ ions, producing solid oxychlorides, reactions (2) and (3). These oxychlorides, when treated with ammonium hydroxide, produce antimony oxide, (4) and (5).
Sb_2_S_3_ + 6HCl → 2SbCl_3_ + 3H_2_S(1)
SbCl_3_ + H_2_O → SbOCl + 2HCl(2)
4SbCl_3_ + 5H_2_O → Sb_4_O_5_Cl_2_ + 10HCl(3)
2SbOCl + 2NH_4_OH → Sb_2_O_3_ + 2NH_4_Cl + H_2_O(4)
Sb_4_O_5_Cl_2_ + 2NH_4_OH → 2Sb_2_O_3_ + 2NH_4_Cl + H_2_O(5)

Hydrolysis of metal chlorides involves various processes and reactions that depend mainly on metal concentration, pH, and chloride concentration [17]. However, the presence of solid intermediate products can make the hydrolysis of antimony trichloride a particularly complex process. Therefore, researchers [18,19] have developed mathematical models and predominance diagrams to predict the behavior of the Sb (III)-Cl-H_2_O system as a function of the aforementioned variables. Experimental validation of theoretical models under specific conditions and experimental studies have been published [20,21].

According to theoretical simulations carried out by Zhao [18], there is a notable correlation between the reduction of chloride concentration from 3 to 1 mol/L and the decrease in hydrolysis pH. This trend is particularly significant within the pH range of 0–4, where oxychlorides remain stable, while above pH 4, oxide becomes more stable and chloride concentration has an insignificant effect. Research conducted by Tian [19] through mathematical modeling and experimental tests on varied chloride concentrations ranging from 3–5 mol/L, had consistent observations with similar trends previously revealed in theoretical analyses. Figure 1 illustrates the precipitated species that were identified at various hydrolysis pH levels.

Although theoretical studies have found that SbOCl is the stable phase within the pH range of −2 to 0, none of the empirical studies have detected this compound in its basic form. The reason for this occurrence is related to the compound instability, leading to a spontaneous transformation into Sb_4_O_5_Cl_2_ [19]. This compound has been obtained in most experimental studies and is typically present within a range of pH values, commonly above pH −0.1 to pH 7 [19,21] with variations depending on antimony and chloride concentrations. There are several distinct crystalline allotropic forms of this compound that have been observed. Other compounds such as Sb_8_O_10_(OH)_2_Cl_2_ have also been identified within pH range between 1.5 to 4.2 [20]. Once at a certain level, it then transforms into Sb_2_O_3_ when the solution reaches a certain pH value (>4.5). Another species detected during similar pH ranges was Sb_8_O_11_C_l2_, which appears at levels ranging from acidity to neutral states [19]. 

The species identified in experimental studies carried out under conditions similar to those studied in this work are shown in Figure 2.

In other studies [22], comparable phases were recognized when using an ethyleneglycol or ethanol and water mixture as the solvent instead of HCl. Sb_4_O_5_Cl_2_ was detected within pH levels ranging from 1 to 2, while Sb_8_O_11_Cl_2_ was identified at values between 4 and 5. Similarly, the presence of Sb_2_O_3_ became apparent within a pH range of approximately 8 to 9. Accordingly, researchers in [23] generate Sb_4_O_5_Cl_2_ by adding water into an ethanol-based HCl solution with a concentration of 2 mol/L. Total precipitation is determined to occur in a volumetric ratio of 1 between the added water and original solution.

It appears that the use of hydrolysis for antimony recovery is feasible in circumstances that closely resemble those present within elution-generated streams. However, more research is needed to fully understand and optimize the process parameters under specific conditions. This work presents findings from experiments carried out to extract antimony via direct hydrolysis in the eluate with water, NaOH, and NH_4_OH. Additionally, product characterization was performed to confirm procedural validity and assess the impact of pH and other metals present in the eluate on extraction yield for antimony. The research aimed to identify favorable conditions that promote maximum antimony extraction efficiency.

## 2. Materials and Methods

### 2.1. Materials

Eluate samples were drawn directly from the Atlantic Copper Industrial Process. Thus, the eluate composition is subject to high dispersion, which is commonly observed in industrial operations. Through a detailed analysis of the historical data provided by Atlantic Copper, a range of values were established that can be considered representative to set the test conditions related to the composition of the eluate. Once these conditions had been established, periodic sampling was performed to obtain eluates with approximate compositions in the range selected. Table 2 shows the concentrations of metals in the eluate sample used in this work.

The reagents used were NH_4_OH and NaOH in analytical grade quality, both provided by Panreac (Barcelona,) Spain.

### 2.2. Methods

The hydrolysis tests were carried out in glass flasks using 50 mL of solution. The flasks were placed in a thermostatic bath to maintain the reaction temperature within ±1 °C, with continuous stirring. Distilled water with conductivity less than 5 µS/cm, NaOH (10% *w/w*), and NH_4_OH (30% *w/w*) were used to modify the pH reaction. After achieving the targeted pH level, the reaction was allowed to proceed for a duration of 60 minutes.. The pH and temperature were continually monitored using a pH meter (Labprocess VioLab PH60. Barcelona, Spain) with a pH accuracy of 0.01. The validity and reliability of the results were ensured by performing a minimum of three tests for each condition.

The precipitated solid was vacuum filtered with 0.65 micron pore membrane filters and dried at 60 °C for 1 h. X-ray diffraction (XRD) patterns of solid dry hydrolyzed products were obtained in θ/2θ geometry using an X’Pert Pro instrument (Malvern Panalytical, Malvern, UK), with Cu-K_α_ radiation source (40 kV, 40 mA). Secondary K_β_ filter and a secondary diffracted beam monochromator were used to reduce fluorescence. XRD patterns were collected by scanning between 0° and 80° in the step-scan mode with 0.03° steps and 5 s dwell time.

Elemental analysis of hydrolysis dry products was carried out by means of X-ray fluorescence (XRF). Previously, the samples were homogenized and pressed on a boric acid tablet. XRF measurements were made on a Zetium Malvern Panalytical wavelength dispersion fluorescence spectrometer (Malvern Panalytical, Malvern, UK) using a semiquantitative measurement method.

The content of Sb, Bi, and As in the liquid phase was measured by inductively coupled plasma optical emission spectroscopy (ICP-OES) on an Agilent 5800 spectrometer (Agilent, Santa Clara, CA, USA).

The concentration of ion chloride in the eluate samples was determined by titration with silver nitrate in an automatic Metrohm 855 tritator (Metrohm, Herisau, Switzerland).

#### Evaluating Speciation and Test Conditions

According to the reference [24], antimony exists in the electrolyte as Sb(III) and Sb(V), with equal distribution, while arsenic is primarily found as As(V) and bismuth exists as Bi(III). Reference [25] suggests that this resin can retain 100% of both Sb(III) and Bi(III), but its capacity for retaining Sb(V) is lower, leading to reduced durability. Thus, it can be concluded that within the eluate, antimony will be present predominantly in a 75–25% ratio between its two oxidation states (Sb III-V), while all bismuth will remain exclusively in the Bi(III) form.

Theoretical studies of the species contained in the eluate have been carried out to limit the study ranges of laboratory tests and mainly refer to arsenic, bismuth, and antimony compounds. However, the accuracy of the results depends mainly on the availability of thermodynamic data of the species involved in the process, and usually experimental tests are required to validate the simulation results. Specifically, the modeling of the Sb-Cl-H_2_O and Bi-Cl-H_2_O systems is complex due to the elevated number of equilibrium reactions involved and the lack of data for some of the solid species that are generated during hydrolysis. However, by using software that employs computational algorithms based on the method of minimizing free energy, it is possible to obtain an estimate of the pH values at which solid species begin to be identified.

To enhance the precision of Sb-Cl classification, the data reported in [26] are implemented. The results for the eluate’s nominal conditions of the evaluation can be found in Figure 3a, which illustrates that solid phase formation takes place at a pH value of 0.3 or lower, where dominant species include SbCl_6_^−3^, SbCl_5_^−2^, SbCl^−4^, and SbCl_3_.

A comparable examination for bismuth can be conducted, whose concentration in the eluate is similar to that of antimony and experiences hydrolysis in a chloride medium [27]. According to Figure 3b, under conditions reflecting nominal composition of the eluate, bismuth oxychloride appears to be highly stable across a wide pH range. The molar fraction versus pH diagram has been derived utilizing equilibrium constants reported by [28] for the Bi-Cl system. As indicated in Figure 3b, precipitation commences around a pH value of 2; below this level bivalent BiCl_6_^3−^ dominates while trivalent BiCl_5_^2−^ appears alongside minor amounts of tetravalent species BiCl_4_. Similarly exhaustive results on bismuth hydrothermal chemistry have also been documented in [29]. In terms of As(V), precipitation is not expected to occur under the eluate conditions. Figure 3c, along with the literature review by [30], indicates that H_3_AsO_4_ and H_2_AsO_4_^−^ are formed by As(V) at varying pH levels.

The pH range selected in this study was determined by reviewing pertinent literature and analyzing the results from speciation diagrams. Specifically, the investigation focused on a range of pH values from 0.25 to 1 as regards antimony extraction efficacy, which is defined here as the proportion between the Sb concentration found in liquid samples through ICP analysis and Sb content measured through the XRF assessment of solid products generated during experimentation according to Equation (6).
Sb yield extraction % = (Sb content in hydrolysis product (g))/(Sb content in solution (g) × 100)(6)

## 3. Results

### 3.1. Effect of pH on Antimony Extraction Yield

The hydrolysis tests were carried out at 25 °C. In the tests performed with NaOH and NH_4_OH, the addition speed was intentionally slow, 1 mL/min, and 0.16 mL/min, respectively, to avoid any increase in temperature resulting from the significant exothermic reaction. However, water addition does not present this constraint; thus, all the necessary volume needed to achieve the specified pH level can be directly incorporated into the eluate.

The results of antimony extraction under different pH conditions are shown in Figure 4. The data reveal that the yield of antimony extraction by hydrolysis is higher when performed with NaOH or NH_4_OH compared to H_2_O. Furthermore, it was observed that the degree of success in extracting antimony depended significantly on pH values; a positive correlation between pH elevation and an increase in yield efficiency can be inferred from the results presented.

In terms of the reagents employed, NH_4_OH exhibited higher values for antimony extraction at all the pH levels tested, with most cases producing an extraction rate surpassing 90%. Additionally, the results displayed favorable reproducibility, as evidenced by low standard deviation values.

Conversely, the use of H_2_O and NaOH as hydrolysis agents produced diminished antimony extraction efficiencies, particularly under low pH conditions. Furthermore, compared to NH_4_OH extractions, extractions using water and NaOH exhibited elevated standard deviation values. Overall, the results indicate that hydrolysis performed with NH_4_OH is the most efficacious approach for antimony extraction in this research study.

In the analysis of results, it is crucial to consider the quantity of reagent required to attain a particular pH level. The volumetric correlations between the volume of eluate and the amount of reagent are shown in the Figure 5. A discernible distinction can be observed among NaOH, NH_4_OH, and H_2_O reagents. The utilization of NH_4_OH results in the lowest volumetric ratio, which fluctuates from 0.48 to 0.55 and requires a 5% increase between various points. In contrast, when NaOH is used, the volumetric ratios escalate to values ranging between 1.7 and 2.3 with changes ranging from 10–15%. To achieve identical pH values, water quantities exceeding a volumetric ratio of 10 are necessary. The outcomes remain reliable despite variations in reagent concentrations and hydroxyl group availability therein.

According to the analysis conducted on the quantity of the reagent, it is clear that the use of HN_4_OH and NaOH provides an advantage over water. However, there are other factors to be considered when comparing all conditions. The use of highly alkaline reagents such as NH_4_OH requires a slow addition rate due to the reaction’s high exothermicity which extends the total reaction time. This phenomenon also occurs in the case of NaOH, though not as extensively as with NH_4_OH; however, this issue does not arise with water whose addition can occur almost instantaneously. Additionally, several important factors such as gas production during reactions, handling, and storage complexities, along with the cost associated with NH_4_HOH and NaOH, must influence our final choice regarding selection of a reagent, but their thorough examination requires further study.

### 3.2. XRD

The diffraction patterns of the solid products resulting from hydrolysis were obtained for all tested conditions. Figure 6 displays the results of X-ray diffraction experiments conducted with water. It can be noted that most of the entities are primarily amorphous, except those obtained at pH 1, where BiOCl (PDF 01-073-2060) featuring a tetragonal configuration was identified. The creation of this compound is attributed to bismuth hydrolysis, and it can occur partially or complete with antimony hydrolysis [28]. However, the results observed at pH 0.25, 0.5, and 0.75 along with those found at pH 1, show discrepancies from the findings reported in the existing literature and the thermodynamic models created based on the eluate conditions. In the case of solids obtained with NaOH and NH_4_OH, as can be seen in Figure 7, they do not have a crystalline character, even at pH = 1, where no hydrolysis products of bismuth are detected as in the case of the assays carried out with water. In some of the products obtained with NaOH, low-intensity peaks of NaCl are identified which appear to be incorporated into the product. The discrepancies encountered in the comparative literature can potentially be ascribed to the existence of arsenic and bismuth, which may result in the lack of crystallinity. Studies conducted on copper electrolytes have extensively described the formation of amorphous antimony and bismuth arsenates [31]. This phenomenon may occur under these evaluation conditions, however, further investigation is required to systematically study and confirm this hypothesis.

### 3.3. Chemical Composition XRF

According to the analysis conducted through crystallography, a significant proportion of hydrolysis by-products lack crystalline structure. Consequently, investigating such samples presents substantial difficulties, especially when trying to understand the behaviour of individual components during a hydrolysis reaction.

Table 3 presents the chemical compositions of the hydrolysis products, which display stoichiometric relationships according to antimony oxychloride compounds. Nevertheless, due to notable quantities of arsenic present within these products and the lack of information of the specific phase, it is challenging to establish an unequivocal identification of each compound. Typically, water-derived products exhibit a higher concentration of antimony content compared to those obtained through the NaOH or NH_4_OH methods which follow in descending order, respectively. The chlorine and oxygen exhibit comparable product compositions similar to those of water and NaOH. However, the implementation of NH_4_OH appears to generate species that could be combinations of oxides and oxychlorides. This is supported by the increase in the percentage of oxygen and the decrease in chlorine.

The inadvertent occurrence of arsenic is of great concern, particularly as it is found in solids at levels approaching 11% when water and NaOH are involved, with values exceeding 20% in those tests involving NH_4_OH. This poses a significant challenge for evaluating the effectiveness of the process and assessing product quality. In all cases, measurable amounts of bismuth are observed that range from 1.5% to approximately 3%. Experiments conducted with NaOH result in final product compositions containing around 2% NaOH.

### 3.4. Effect of pH on Antimony, Bismuth, and Arsenic Content

A study was conducted on the effect of pH on the composition of the main elements. In Figure 8, the evolution of the Sb content with pH is observed using different reagents. At pH 0.25, there is a significant increase in Sb content (68.38%) within the solid powder produced by hydrolysis using water as the medium. However, with NaOH or NH_4_OH serving as alternative hydrolysis agents, there was a decrease in Sb concentration that resulted in values of 59.65% and 53.89%, respectively. Similar observations can be made at pH levels of 0.5 and 0.6, whereby the percentage of antimony in solid powder derived from water hydrolysis exceeds that obtained by NaOH or NH_4_OH hydrolysis methods. However, at a pH level of 0.75, the percentage yield is comparable between Sb extracted through water and NaOH hydrolysis approaches (at 61.79%), while the use of NH_4_OH results in marginally lower yields (53.95%). At pH 1, the percentage of Sb in solid powder obtained through hydrolysis with water is the lowest (36.56%). Interestingly, the percentage of Sb obtained with NaOH and NH_4_OH is higher (58.86% and 59.04%, respectively), suggesting that these two hydrolysis media are more effective than water for extracting Sb at this pH.

The results of evolution of bismuth are shown in Figure 9. Bismuth values remain relatively constant in the solids produced using NaOH and NH_4_OH with percentages below 2%. On the contrary, when H_2_O is used as a reagent for solid production, values exceed this threshold across all pH levels ranging from 0.25 to 0.75, reaching 5% at their maximum value. A substantial rise in the amount of Bi is observed at a pH level of 1. It could potentially be attributed to the initiation of the bismuth hydrolysis reaction, which exists in amounts equivalent to antimony. This is substantiated by the identification of BiOCl according to X-ray diffraction outcomes.

Figure 10 shows the percentage composition of arsenic in solid powders that were hydrolyzed with water, NaOH, and NH_4_OH at various pH levels. The results indicate that at a pH of 0.25, the arsenic content in the powder obtained by hydrolysis with H_2_O is lower than that obtained by using NaOH and NH_4_OH (8.57%, 12.18%, and 23.51%, respectively). As the pH increased with time, a decrease in arsenic concentration occurred within the powders acquired by hydrolysis with water; however, there was no significant change for those produced using NaOH or NH_4_OH solutions which remained constant throughout each analyzed test phase (pH = 0.75–10.36%, As-H_2_O vs. 13.09%, As-NaOH vs. 22.06% As-NH_4_OH). At pH 1, a decrease in solids obtained with H_2_O and NH_4_OH is observed.

## 4. Conclusions

Regarding antimony yield extraction, the results of this study demonstrate that the efficacy of antimony extraction by hydrolysis was higher when performed using NH_4_OH or NaOH compared to water. Additionally, it was observed that pH values significantly impact the degree of success in the extraction of antimony. According to the results, the optimal conditions are obtained at pH 0.5 for water and pH 1 for NH_4_OH and NaOH, where the average results of antimony extraction yield were 90.4, 96.1, and 96.7, respectively.Most solids lack a crystalline structure, making it difficult to identify the nature of the compounds formed. Despite this limitation, element concentrations suggest that certain precipitates may contain oxychloride or oxide compounds.Arsenic was generally incorporated into all solids obtained, which affects the purity of the product. In this sense, water is more selective, as the average antimony content was higher (68.38%) and the arsenic values lower (8%) than in the case of NaOH and NH_4_OH.Regarding bismuth, its integration into solids was markedly inferior to arsenic (less than 2%) and remained unaffected by pH levels except in the water test where hydrolysis product of bismuth was identified at a pH of 1. This phenomenon accounts for the reduction observed in antimony extraction yields under such conditions.Although the performance obtained with HN_4_OH was higher than that of the other reagents, the excess arsenic that was incorporated into the solids hinders the potential commercialization of the product. Water is more selective, but the necessary amounts are much larger than those in the case of the other two reagents. NaOH offers the best compromise between performance and selectivity.

## Figures and Tables

**Figure 1 materials-16-03918-f001:**
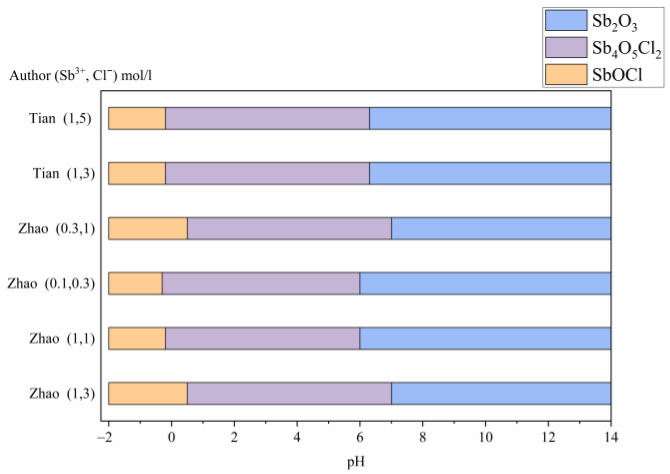
Distribution of species as a function of pH according to theoretical studies [18,19].

**Figure 2 materials-16-03918-f002:**
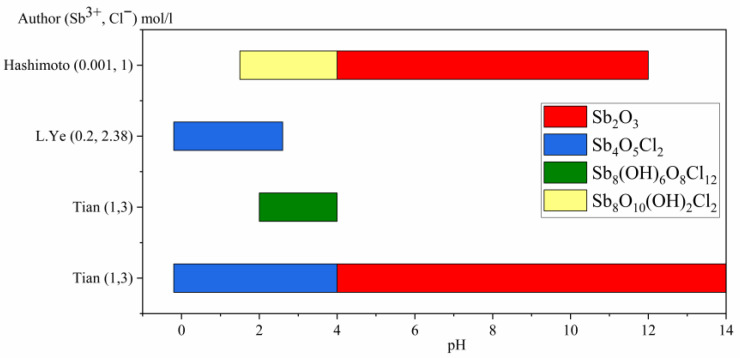
Distribution of species as a function of pH according to experimental studies [19,20,21].

**Figure 3 materials-16-03918-f003:**
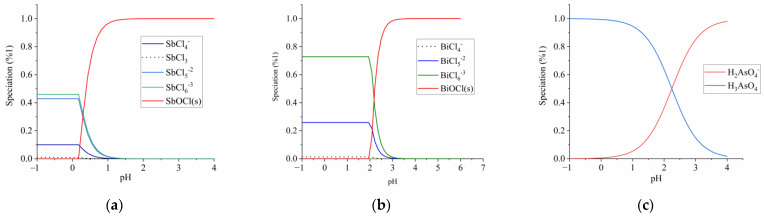
Speciation diagrams in the form of a molar fraction as a function of pH for: (**a**) Sb (III) -Cl-H_2_O system, for a solution containing 4.7 mol/L HCl and 0.08 mol/L Sb. (**b**) Bi(III) -Cl-H_2_O system, for a solution containing 4.7 mol/L HCl and 0.04 mol/L Bi. (**c**) As(V)-Cl-H_2_O system, for a solution containing 4.7 mol/L of HCl and 0.04 mol/L.

**Figure 4 materials-16-03918-f004:**
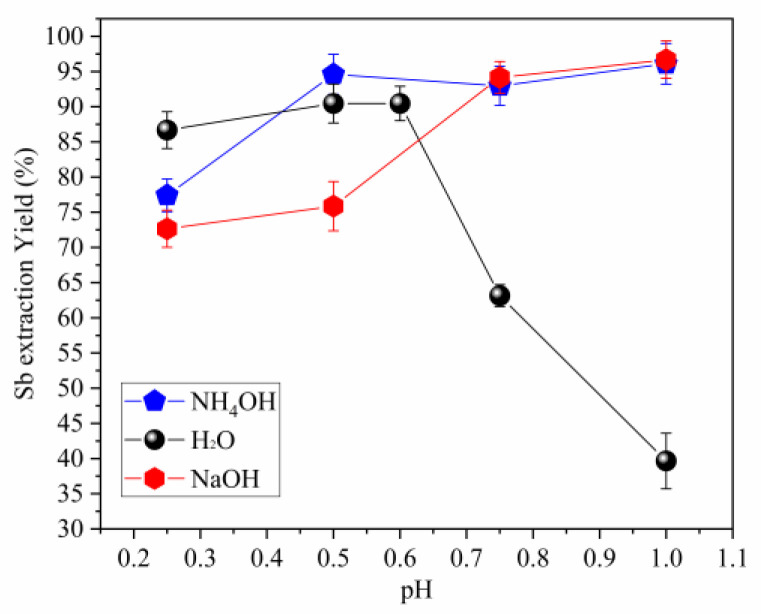
Antimony extraction yields results at 25 °C. (The bar represents the standard deviation.)

**Figure 5 materials-16-03918-f005:**
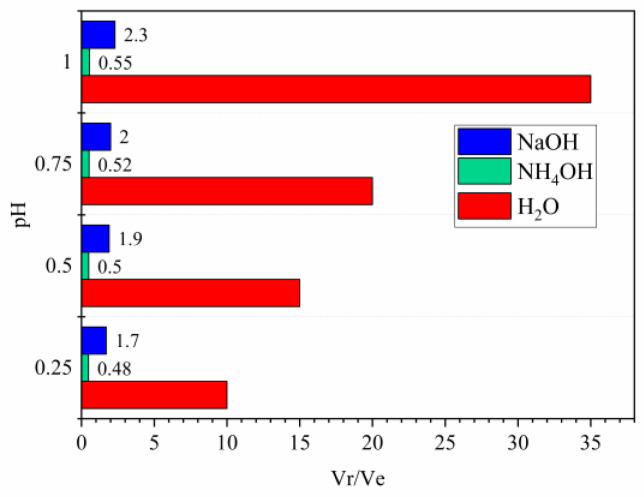
Ratio between the volume of reagent (Vr) used and the volume of eluate sample (Ve) required to reach the pH values indicated.

**Figure 6 materials-16-03918-f006:**
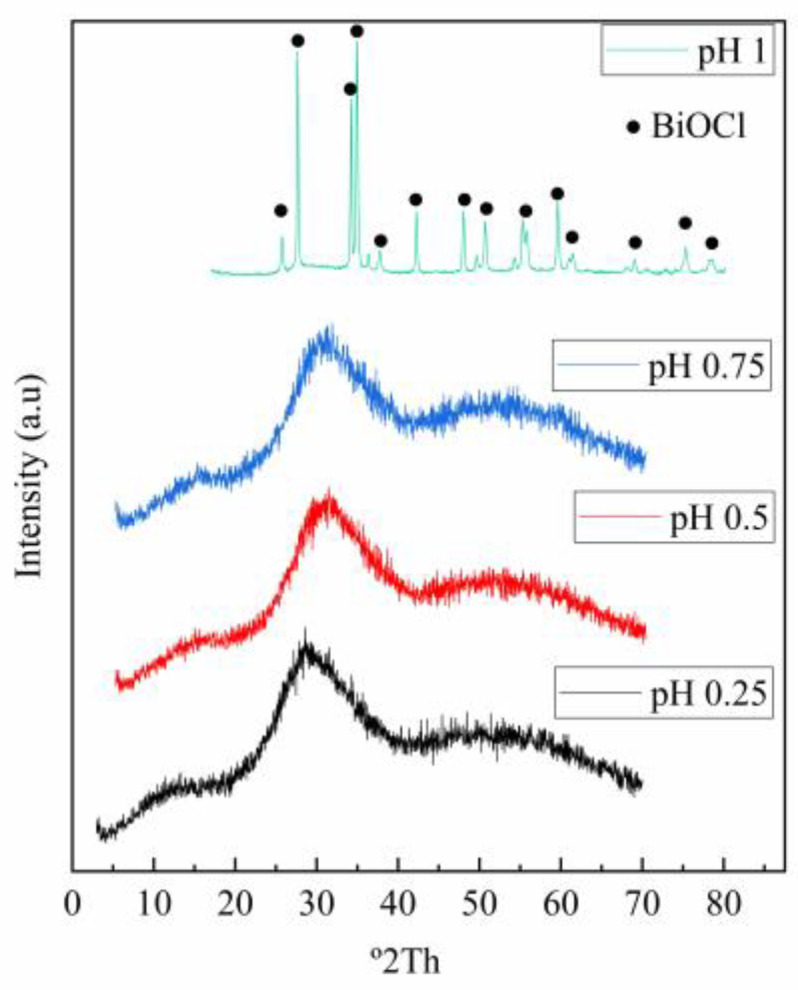
Representative diffractograms obtained for the hydrolysis products with water at different pH and 25 °C.

**Figure 7 materials-16-03918-f007:**
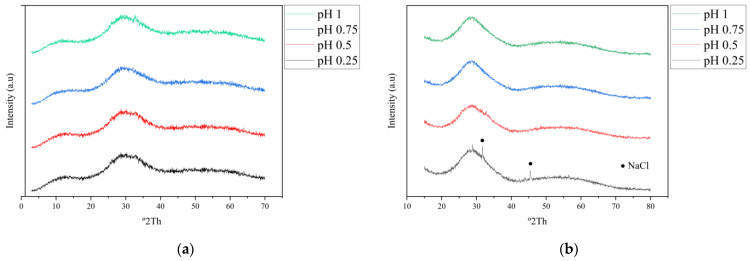
Representative diffractograms obtained for the hydrolysis products at different pH and 25 °C. (**a**) Hydrolysis with NH_4_OH. (**b**) Hydrolysis with NaOH.

**Figure 8 materials-16-03918-f008:**
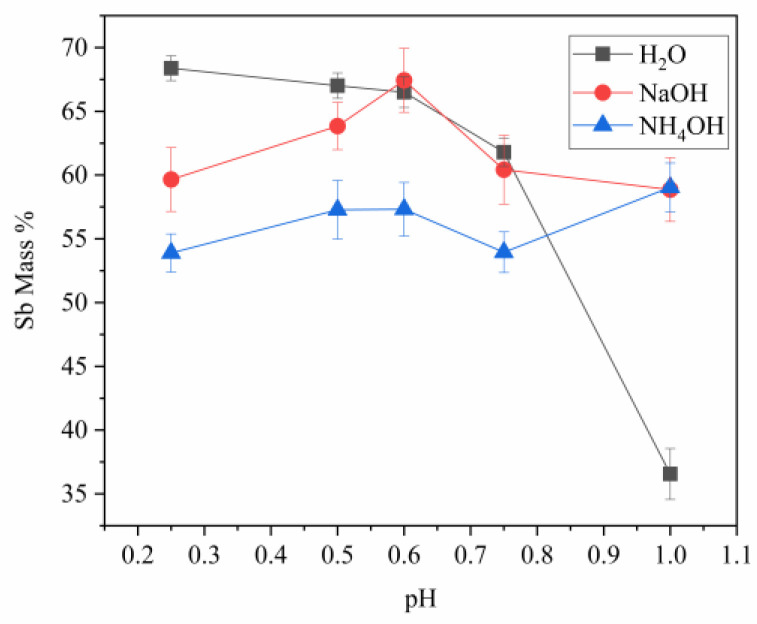
Evolution of the antimony content with pH (bars represent standard deviation).

**Figure 9 materials-16-03918-f009:**
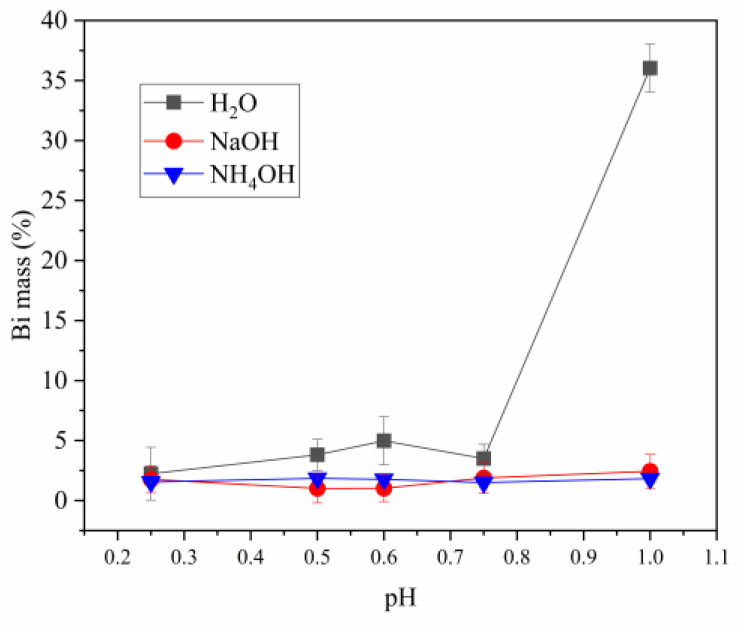
Evolution of the bismuth content with pH (bars represent standard deviation).

**Figure 10 materials-16-03918-f010:**
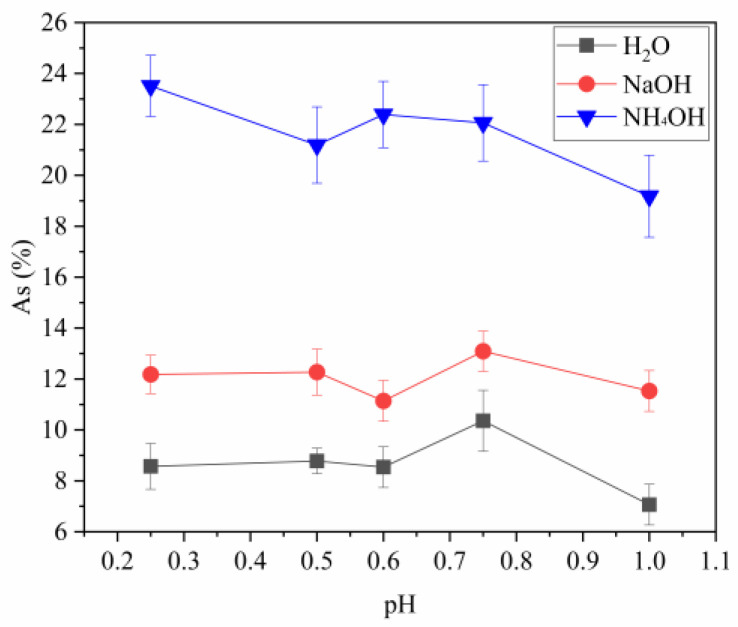
Evolution of arsenic content with pH (bars represent standard deviation).

**Table 1 materials-16-03918-t001:** Typical composition of the eluate (g/L) from the ion exchange facility.

As	Bi	Sb	Cl^−^	Others
2–5	8–10	9–10	170	<1

**Table 2 materials-16-03918-t002:** Composition of the main components of eluate used in this study (g/L).

Element	As	Bi	Sb	Cl^−^
Content (g/L)	2.3	9.5	8.9	170

**Table 3 materials-16-03918-t003:** Mass percent average compositions of the products obtained in the hydrolysis.

Element	Sb	Cl	O	As	Bi	Na	Others
H_2_O	64.75 ± 3.59	4.01 ± 2.46	16.32 ± 1.81	10.90 ± 2.04	2.93 ± 1.73	-	0.13 ± 0.1
NaOH	62.19 ± 3.89	5.20 ± 3.89	17.02 ± 0.45	11.78 ± 0.94	1.49 ± 0.58	1.98 ± 0.79	0.34 ± 0.16
NH_4_OH	56.04 ± 2.55	2.21 ± 1.43	18.32 ± 0.23	21.48 ± 1.81	1.68 ± 0.18	-	0.36 ± 0.11

## Data Availability

The data presented in this study are available upon request from the corresponding author.

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
