# Peer review of "Effect of pH Hydrolysis on the Recovery of Antimony from Spent Electrolytes from Copper Production"

_materials, 2023, doi:10.3390/ma16113918_

Round 1
Reviewer 1 Report
The authors present the effect pH hydrolysis on the recovery process for antimony extracted from the spent electrolytes. This research could give the favorable conditions for promoting maximum antimony efficiency. This manuscript can be accepted after revision.
1. The figures in maunuscript should be modified. most of them are unclear and the texts in figures are to small.
2. The authors mentioned the morphology of the recovered antimony samples, but there is no related data in manuscript. please adding the data, such as SEM images.Author Response
Please see the attachment

Reviewer 2 Report
In this manuscript, the recovery of antimony from spent electrolytes copper production and the effect of pH hydrolysis on antimony flotation were studied. As mentioned in the article, the pH hydrolysis can affect the recovery process for antimony extracted from spent electrolytes. The manuscript explored the optimal pH scheme for antimony extraction through adjustment pH by different OH reagents. The subject considered in this work is interesting and the paper is well organized. However, the submitted manuscript requires further revisions before publication. Please check the attachment.

Moderate editing of English language.
Reviewer 3 Report
This manuscript describes the antimony recovery from spent electrolytes from copper production, especially the pH effect. I strongly agree with the need for research from the point of view of the circular economy. However, the manuscript itself needs further improvement before its publication.
1. The section heading should be re-arranged. The subsection in the introduction (1.1 background) seems unnecessary. Section 3.2 was not presented, while section 3.2.1 was provided.
2. Inconsistent data set: Some data sets did not show a consistent trend. For example, the As dissolution in water in Fig 10, the pH 0.75 showed a sudden increase, but the reason is not well explained. Providing an error bar will be the option to give confidence in the data set.
3. Economical analysis: Since it is about circular economy, economic analysis will be an important point to show in this paper. Moreover, the carbon footprint can also be considered.
Minor editing of English language required
Round 2
Reviewer 3 Report
The manuscript has been revised according to the reviewer's comment. I do not have any further comments on this manuscript.
Minor editing of English language required